

# Relationship between the molecular composition, visible light absorption, and health-related properties of smoldering woodsmoke aerosols

Lam Kam Chan, Khanh Q. Nguyen, Noreen Karim, Yatian Yang[‡], Robert H. Rice, Guochun He, Michael S. Denison, and Tran B. Nguyen*

Department of Environmental Toxicology, University of California Davis, Davis, CA 95616
[‡] *Now at: Department of Biochemistry and Molecular Medicine, UC Davis School of Medicine, Sacramento, USA*

*Correspondence to*: T.B. Nguyen (*tbn@ucdavis.edu*)

**Abstract.** Organic aerosols generated from the smoldering combustion of wood critically impact air quality and health for
billions of people worldwide; yet, the links between the chemical components and the optical or biological effects of
woodsmoke aerosols (WSA) are still poorly understood. In this work, an untargeted analysis of the molecular composition of
smoldering WSA, generated in a controlled environment from nine types of heartwood fuels (African Mahogany, Birch,
Cherry, Maple, Pine, Poplar, Red Oak, Redwood, and Walnut) identified several hundred compounds using gas
chromatography mass spectrometry (GC-MS) and nano-electrospray high-resolution mass spectrometry (HRMS) with tandem
multistage mass spectrometry (MS$^n$). The effects of WSA on cell toxicity, aryl hydrocarbon receptor (AhR)- and estrogen
receptor (ER)-dependent gene expression were characterized with cellular assays, and the visible mass absorption coefficients
(MAC$_{vis}$) of WSA were measured with UV-visible spectroscopy. The WSA studied in this work have significant levels of
biological and toxicological activity, with exposure levels in both an outdoor and indoor environment similar to or greater than
those of other toxicants. A correlation between the HRMS molecular composition and aerosol properties found that phenolic
compounds from the oxidative decomposition of lignin were the main drivers of aerosol effects, while the cellulose
decomposition products played a secondary role, e.g., levoglucosan was anti-correlated with multiple effects. Polycyclic
aromatic hydrocarbons (PAHs) were not expected to form at the combustion temperature in this work, nor were observed
above the detection limit; thus, biological and optical properties of the smoldering WSA are not attributed to PAHs. Syringyl
compounds tend to correlate with cell toxicity, while the more conjugated molecules (including several compounds assigned
to dimers) had higher AhR activity and MAC$_{vis}$. The negative correlation between cell toxicity and AhR activity suggests that
the toxicity of WSA to cells was not mediated by the AhR. Both mass-normalized biological outcomes had a statistically-
significant dependence on the degree of combustion of the wood. In addition, our observations support that the visible light
absorption of WSA is at least partially due to charge transfer effects in aerosols, as previously suggested. Finally, MAC$_{vis}$ had
no correlation with toxicity or receptor signaling, suggesting that key chromophores are not biologically active on the endpoints
tested.



## 1. Introduction

The combustion of wood from, for example, residential fireplaces, forest fires, and prescribed burns is a large source of fine particulate matter ($PM_{2.5}$) in the United States and much of the world (Rogge et al., 1998; EPA, 2003; Mazzoleni et al., 2007; Park et al., 2007; Swiston et al., 2008; Zhang et al., 2013), particularly in winter when woodsmoke aerosols (WSA) can account for the majority of organic carbon and up to 90% of $PM_{2.5}$ regionally (Rogge et al., 1998; Gorin et al., 2006; Kleeman et al., 2008; NYSERDA, 2008). In an indoor environment, the incomplete combustion of solid fuels (64 – 95% wood usage in rural India at, but also dung and crops (Menon, 1988)) is the main source of indoor air pollution exposure to roughly 3 billion people worldwide (WHO, 2011). This indoor WSA exposure occurs mostly in developing nations and mostly to women and children, possibly accounting for the highest burden of environmental disease globally (Ezzati and Kammen, 2002; WHO, 2002; Smith and Mehta, 2003).

Multiple factors impact the combustion chemistry, such as the wood composition (lignin/cellulose fractions, lignin H/G/S composition, natural monomers such as resins, waxes, sugars), water fraction, environmental conditions, and other factors, which in turn impact the aerosol composition and effects. During a typical fire, the high-intensity but short initial period of flaming combustion is correlated with $CO_2$ production and emissions that rise to higher altitudes. In contrast, the low-intensity but long period of smoldering combustion is correlated with CO and other products of incomplete combustion, and emissions that remain closer to the ground (Andreae and Merlet, 2001) where they are more likely inhaled. Smoldering combustion, which produces a higher quantity of particles and most of the organic species in a fire (Einfeld et al., 1991; Yokelson et al., 1997; Bertschi et al., 2003), is responsible for a majority of biomass consumption during prescribed burns, residential wood burning, cigarette smoke, and fires from tropical and temperate forests and deforestation areas (Standley and Simoneit, 1990; Ward et al., 1992; Yokelson et al., 1997; Simoneit et al., 2000; Ohlemiller, 2002; Rabelo et al., 2004). Due to the high toxicant production, the emissions from smoldering combustion are recommended to be used to assess health exposure (Morawska and Zhang, 2002). This work focuses on the organic aerosols generated from smoldering combustion of woods in a controlled laboratory setting.

The significant adverse health effects of WSA on endpoints such as mortality, morbidity, respiratory disease, cancer, among others, are well-documented and are characterized to be as serious, if not more so, than other sources of particulate matter (Lewis et al., 1988; Zelikoff et al., 2002; Boman et al., 2003; Kubátová et al., 2006; Orozco-Levi et al., 2006; Lewtas, 2007; Swiston et al., 2008; Danielsen et al., 2011). For example, organic extracts of particulate matter with high WSA content are approximately 30 times more potent than cigarette smoke condensate in a tumor induction assay (Cupitt et al., 1994). Yet, it is not clear whether the health-related effects of the WSA organics can be attributed to toxicants other than the well-studied polycyclic aromatic hydrocarbons (PAHs), such as the multifunctional aldehydes, ketones, phenols, organic nitrogen, organic acids, etc. (Hedberg et al., 2002) that are also present in WSA with largely unknown biological effects. Previous bulk analyses have shown that the emissions of total particulate matter and PAHs depend on the wood type and burn conditions (Maga and



Chen, 1985; Bølling et al., 2009; Nyström et al., 2017). High-resolution mass spectrometry (HRMS) analyses have offered insight into the complex molecular composition of aerosols from biomass combustion (Smith et al., 2008; Laskin et al., 2009; Lin et al., 2016; Lin et al., 2017; Fleming et al., 2018). However, the links between the chemical origin and macroscopic endpoints are still poorly understood. This work focuses on two critical gaps in knowledge surrounding woodsmoke (Naeher et al., 2007): the fundamental understanding of (1) how different types of woods and burn conditions affect the molecular composition and (2) relationships between the aerosol's properties and its chemical composition.

Alongside biological effects, we also studied the potential chromophores responsible for the visible light absorption of smoldering WSA. The combustion of wood (and other biomass) is known to produce "brown carbon" aerosols (Andreae and Gelencser, 2006), which absorb light in the troposphere and appear yellow or brown to the eye. Although most of the absorption of WSA and other biomass burning aerosols is due to elemental carbon (soot) in the atmosphere, the organic aerosols such as in smoldering WSA represent a secondary but not insignificant fraction (Kirchstetter and Thatcher, 2012; Washenfelder et al., 2015).

## 2. Experimental

### 2.1. Smoldering combustion woodsmoke aerosols

Woodsmoke aerosol (WSA) samples were prepared in a 40 L smoldering combustion chamber (**Fig. 1**) that is fitted with Teflon tubing. The relative humidity (RH) for the combustion was set to ~ 40%, as verified by a membrane humidity probe (Vaisala Inc.), chosen to be consistent with the RH range found in some indoor environments (Salonvaara et al., 2004; Irulegi et al., 2017), where residential wood burning may occur, and with the daytime RH that are correlated with wildfire events (highest burned areas at RH 38 - 42%) (Flannigan and Harrington, 1988; Piñol et al., 1998). Humidified ultra zero air in the chamber was achieved by combining a ~ 1.5 L min⁻¹ humid flow through a wet-wall humidifier with a ~ 2.5 L min⁻¹ dry flow from a high-pressure cylinder (Airgas UZA: 20 – 22% $O_2$ in $N_2$, 2 ppm $H_2O$, 1 ppm $CO_2$, 1 ppm CO, 0.1 ppm total hydrocarbons) through flow restrictors.

Heartwood samples of nine types of wood (African Mahogany, Birch, Cherry, Maple, Pine, Poplar, Red Oak, Redwood, and Walnut) were purchased from a lumber store local to Davis, CA and were cut into 4 cm x 2 cm x 1 cm blocks, using a blade cleaned with isopropanol between each use. The subset of wood species selected are a combination of softwoods (Redwood, Pine) and hardwoods that can be found in North American forests (Burns and Honkala, 1990) and often used for firewood or for building homes and furniture. Approximately two wood blocks with an average mass of 2.5 g each were used for every burn with a combustion temperature of 330 °C, which produced a thick white smoke from the smoldering process. After 15 minutes of combustion, aerosols were collected from the center of the chamber, approximately 15 – 20 cm above the fuel source, for 25 minutes with a Polyvinylidene fluoride filter (Millipore Sigma, 0.65 μm pore size) and a flow rate of





approximately 2 L min$^{-1}$ (the remaining flow is discarded through the vent). An average of 85% by mass of the fuel source

was burned. The temperature at the location immediately upstream of aerosol collection was approximately 25 °C. The tubing

in contact with aerosols and filter collection apparatus was cleaned between each burn using isopropanol. Multiple (> 3) burns

of each wood type were performed, with reproducible results.

The WSA samples were gravimetrically analyzed and, if not used immediately, vacuum-sealed in polyethylene filter holders

and stored at –10°C. Prior to analyses, WSA were extracted in solvent using a mixture of 1:1 ultrapure water and acetonitrile

with 30 minutes room-temperature agitation on an orbital shaker (IKA Inc.) at 1000 rpm. Each extraction used the required

solvent to achieve a mass concentration of 5 mg mL$^{-1}$, depending on the WSA mass collected, and were diluted for analyses

as necessary.

## 2.2. Molecular composition analysis

Diluted WSA extracts (100 μg mL$^{-1}$) were analyzed for organic molecular composition using a linear-trap-quadrupole Orbitrap

(LTQ-Orbitrap) mass spectrometer (Thermo Corp.) at a mass resolving power of ~ 100,000 m/Δm at m/z 200. The WSA

extracts were directly infused into a capillary nano-electrospray ion source (ESI, 50 μm fused-silica capillary tip, 4 kV spray

voltage, 275°C capillary temperature, 15 μL min$^{-1}$ flow), and the spectra were taken at a mass range of m/z 50 – 1000 in the

positive ion mode. Ionization mechanisms in the positive ion mode ESI include protonation ([M+H]$^+$ ions) and/or sodiation

([M+Na]$^+$ ions). An external 11-point mass calibration in the m/z range of 100 – 600 was performed using a variety of analytical

standard solutions (ESI-L tuning mix, amine mixture, and others, (Supelco Inc.)) immediately prior to the MS analysis, such

that the mass accuracy is adjusted to be approximately 1 ppm for standard compounds. Insights into molecular structure were

obtained using collision induced dissociation (CID) multistage tandem mass spectrometry (MS$^n$, stages 2 – 4) in the LTQ-

Orbitrap for ions that have adequate signal. CID energy was tuned for each mass, so that the normalized precursor ion had 10

– 20% abundance, and neutral or radical losses were analyzed using the Thermo Xcalibur software suite.

The sample mass spectra with signal to noise ratio (S/N) > 3 were processed by subtracting the background mass spectra of

the blank filter extracts, deconvoluted with a quadratic fit model and deisotoped using Decon2LS tools (freeware from PNNL),

mass corrected with the external calibration curve, and assigned to molecular formulas using a custom Matlab protocol based

on heuristic mass filtering rules (Kind and Fiehn, 2007) and Kendrick Mass (KM) defect analysis (Kendrick, 1963; Roach et

al., 2011) with KM base of $CH_2$. Correlations with HRMS peaks are performed using Matlab's linear regression model fitlm.

Least-squares fit results with coefficients ($R^2$) greater than 0.4 having slopes in either direction are reported.

The less polar, higher-volatility components of WSA were extracted using ethyl acetate at 100 μg mL$^{-1}$ and analyzed with gas

chromatography mass spectrometry (GC-MS) on an Agilent 6890 GC and 5973 MS detector using an HP-5MS column (20 m

x 250 μm x 0.25 μm) and the following temperature program: 50°C (0.5 min hold), 12°C min$^{-1}$ until 260°C (12 min hold). An

EPA-610 method for PAHs (US EPA, 2005) and PAH certified calibration standard (TraceCERT, 16 components) were also



used to verify if the WSA contained PAHs. The analytes were identified by their electron-ionization fragmentation at 70 eV – the spectra were compared to those from the GC-MS library from the National Institute of Standards and Technology.

### 2.3. Mass Absorption Coefficient

The visible light absorption of WSA extracts were measured using a dual beam UV-visible spectrophotometer (UV-1800, Shimadzu Corp.) and 1 cm quartz cuvettes at multiple extract concentrations. The per-gram light absorption of WSA extracts
were calculated as a mass absorption coefficient (Moosmüller et al., 2011):

$$MAC(\lambda) = \frac{A(\lambda)*\ln(10)}{C*L} \tag{1}$$

where $A(\lambda)$ is the wavelength-dependent absorbance of the WSA extract, C is the extract concentration (0.005 g mL$^{-1}$), and L is the pathlength of the light absorption (1 cm). To account for the wavelength dependence of MAC and compare between the WSA extracts in this study, we used an average MAC that was integrated in the visible region:

$$MAC_{vis} = \frac{1}{700nm-400\,nm} \int_{400\,nm}^{700\,nm} MAC(\lambda)d\lambda \tag{2}$$

The absorption spectra (**Fig. S1**) of the WSA extracts in this study are consistent with other ambient brown carbon spectra, where a featureless tailing absorption is observed into the visible wavelengths. MAC$_{vis}$ values determined at multiple extract concentrations varied less than 5% (**Fig. S2**).

### 2.4. Cell Toxicity Bioassay

Human epidermal keratinocytes were used to determine the toxicity of WSA. When exposed to toxic xenobiotic compounds, keratinocytes can display a feature of terminal differentiation, formation of an envelope of isopeptide cross-linked protein at the plasma membrane (Rice and Green, 1979). The cross-linking can be induced by permeabilization of the cells, permitting a rise in cytosolic calcium ions that activates the membrane-bound transglutaminase TGM1 and results in cell death. The protein content from envelope formation serves as a quantitative biomarker of cell damage from toxic exposures.

Briefly, cells were cultured and assayed as follows: spontaneously immortalized keratinocytes (SIK, passages 25-35) were serially cultivated in a 2:1 mixture of Dulbecco-Vogt Eagle's and Ham's F-12 media supplemented with fetal bovine serum (FBS) (5%), hydrocortisone (0.4 µg/ml), epidermal growth factor (10 ng/ml), insulin (5 µg/ml), transferrin (5 µg/ml) and 0.18 mM adenine (Allen-Hoffmann and Rheinwald, 1984; Rice et al., 1993). After reaching confluence in 24 well plates, the cell cultures were incubated in a medium lacking epidermal growth factor for two days. Cultures were then incubated for 48 h with
extracts of WSA (5 mg/mL in 1:1 v/v water:acetonitrile, 15 – 240 µL) in the concentration range of 0.23 to 3.75 mg/mL in a total well volume of 320 µL. After incubation, the medium was discarded, and the cultures were further incubated for 24 h in an aqueous solution of 2% sodium dodecyl sulfate, 20 mM dithiothreitol and 50 mM Tris buffer (pH 8) at 37ºC. The envelopes



from each well were isolated by centrifugation, rinsed 3 times in 0.1 % sodium dodecyl sulfate and assayed for relative protein content with bicinchoninic acid, quantified colorimetrically at a wavelength of 495 nm (Smith et al., 1985). WSA extracts were

assayed at each concentration in triplicate. Blank controls (extracted blank filter) and solvent controls did not result in envelope formation, nor inhibition of envelope formation.

The half maximal effective concentration ($EC_{50}$) of the WSA extract that induced a cell toxicity response halfway between the baseline and the maximum activity, was obtained using concentration-response curves fitted in Origin software with adjusted $R^2 > 0.94$. The $EC_{50}$ uncertainties were propagated from fit uncertainties and the standard deviations of triplicate analyses.

Individual concentration-response curves and fits for the WSA extracts are shown in **Figure S3**.

## 2.4. Aryl hydrocarbon Receptor (AhR) and Estrogen Receptor (ER) bioassays

Screening of WSA extracts for AhR- or ER-active compounds were carried out using recombinant mouse hepatoma (H1L6.1c2) cells containing a stably integrated AhR-responsive luciferase reporter gene plasmid (pGudLuc6.1) and human breast carcinoma (VM7Luc4E2) cells containing a stably integrated ER-responsive luciferase reporter gene plasmid

(pGudLuc7ere), as previously described (He et al., 2014; Brennan et al., 2016). Briefly, for AhR analysis, H1L6.1c2 cells in growth medium (alpha minimal essential media (aMEM) containing 10% FBS) were plated (75,000 cells/well) into white, clear-bottomed 96-well tissue culture plates and incubated at 37°C for 24 h prior to the addition of WSA extracts. For ER analysis, VM7Luc4E2 cells were switched from maintenance medium (aMEM containing 10% FBS) to estrogen-stripped medium (phenol red-free aMEM containing 10% charcoal-stripped FBS), incubated for 3 days at 37°C before plating into

white, clear-bottomed 96-well tissue culture plates at a density of 75,000 cells/well and further incubated at 37 °C for 24 h prior to chemical addition. Cells were incubated with 1 μL of: DMSO solvent (1% (v/v)), the reference standard dissolved in DMSO (2,3,7,8-tetrachlorodibenzo-p-dioxin (TCDD) for H1L6.1.c2 cells and 17β-estradiol (E2) for VM7Luc4E2 cells), acetonitrile solvent, or WSA extract for 24 h.

After incubation, cells were visually inspected for signs of toxicity, rinsed twice with phosphate-buffered saline, followed by

the addition of Promega cell lysis buffer, and shaking for 20 min at room temperature to allow complete cell lysis. Luciferase activity in each well was measured by using an Orion microplate luminometer as previously described (He et al., 2014). Luciferase activity was corrected for background activity in DMSO-/acetonitrile-treated cells and values (in relative light units (RLUs)) expressed as a percent of the luciferase activity obtained with the maximally inducing concentration of TCDD or E2. All incubations and analyses were performed in triplicate. TCDD bioanalytical equivalents (BEQs) for AhR active samples

were determined by comparison of the luciferase activity of the sample extract to that in a TCDD concentration response curve, as described elsewhere (Baston and Denison, 2011), and shown in **Figure S4**. The relative magnitude of AhR and ER induction by WSA (5 x $10^{-6}$ g of original material in each plate well) was normalized to the equivalent level of AhR and ER luciferase gene induction from the TCDD and ER concentration response curves, respectively, on a g/g basis. The uncertainties in AhR



and ER relative activity were propagated from the uncertainties in the triplicate determinations and the reference calibration
concentration-response curves.

## 3. Results and Discussion

### 3.1. Biological and optical properties of WSA

**Table 1** shows cell toxicity, AhR activity, ER activity and $MAC_{vis}$ results for the nine types of WSA extracts studied in this
work, normalized by the mass of the original aerosol material. The results provide a clear indication that this type of complex
mixture can affect macromolecular targets in cells. An important question is how relevant are the findings with smoldering
WSA to health effects from actual exposures. Considering toxicity in an outdoor environment such as a wildfire event, $PM_{2.5}$
concentrations in excess of 300 μg/m$^3$ can be observed in surrounding areas (Wu et al., 2006; Wegesser et al., 2009) and up to
2.7 mg/m$^3$ near the burn site (Adetona et al., 2011). As much of the aerosol comes from the first and second smoldering phases
(Alves et al., 2011), a realistic exposure to smoldering WSA in cities surrounding a fire can be 100 μg/m$^3$. Assuming a typical
pulmonary tidal volume of 500 mL for adults (Beardsell et al., 2009), and the cell toxicity $EC_{50}$ of Red Oak smoldering WSA,
we estimate that a human adult would take ~ 2175 breaths to reach the $EC_{50}$ value. As a normal respiratory rate for an adult at
rest is 12 breaths per minute (Carey et al., 2005), the $EC_{50}$ limit is reached after 3 h of woodsmoke exposure, yet residents near
fires can be exposed to the smoke for many days (Wegesser et al., 2009). The acute exposure health risk of residential wood
burning in an indoor environment is more severe. $PM_{2.5}$ concentrations around cookstoves are high (e.g., 1.8 mg/m$^3$)
(Chowdhury et al., 2012), such that the $EC_{50}$ limit is reached in ~ 120 breaths or 10 minutes, a highly plausible exposure
duration that occurs almost daily for some people.

Considering the AhR activity of WSA, the interpretation is more complex. AhR (Denison and Heath-Pagliuso, 1998) is a
ligand-activated transcription factor that mediates the biochemical and toxic effects of various well-recognized environmental
toxicants, such as dioxins (Mandal, 2005; Denison et al., 2011) and PAHs (Machala et al., 2001; Barron et al., 2004; Billiard
et al., 2006), but also a wide variety of other structurally-diverse chemicals (Denison and Nagy, 2003; Nguyen and Bradfield,
2007; Petkov et al., 2010). However, AhR-dependent (i.e. dioxin-like) toxicity appears to be mediated only by dioxin-like
chemicals (e.g., polyhalogenated dibenzo-p-dioxins, dibenzofurans and biphenyls) and not all AhR agonists, the majority of
which are metabolically inactivated. Determination of the presence of toxic AhR ligands in a sample requires extensive sample
workup to isolate dioxin-like chemicals from the large number of nontoxic AhR active chemicals. Thus, the AhR bioassay
results for WSA in this work provide a measure of the overall ability of the extract to activate the AhR and AhR-dependent
gene expression and does not provide a direct measure of AhR-dependent toxic potency. However, they do not rule out the
presence of toxic AhR ligands; further sample isolation and analytical analysis would be needed to confirm the presence of
dioxin-like chemicals in the WSA samples. The relatively high levels of AhR active chemicals observed in the WSA samples
tested here could potentially lead to adverse effects via the ability of these compounds to stimulate AhR-dependent expression


of gene products such as cytochrome P4501A1, which can contribute to the metabolic activation of PAHs into their mutagenic form, as well as contribute to increased oxidative stress as a result of the production of reactive oxygen species as a byproduct of CYP1A1 metabolism of endogenous and exogenous chemicals (Park et al., 1996; Nebert et al., 200, 2004). Our results are also fairly consistent with the previous observation that WSA has a relative carcinogenesis potency of $1 – 6 \times 10^{-4}$ compared to pure benzo(a)pyrene (Lewtas, 2007), another AhR ligand with well-recognized adverse health effects, although the lack of

information about the specific wood fuels and burn conditions used in that study do not allow direct comparisons to the WSA results presented here.

Both the AhR bioassay and cell toxicity results are consistent with epidemiological and toxicological evidence for negative human health impacts of WSA ((Naeher et al., 2007), and references therein). In light of previous findings that smoldering combustion at medium temperature produces particles that are more toxic than those from high-temperature combustion or

complete combustion ((Bølling et al., 2012), and references therein), the results from this work suggest that the smoldering fraction of WSA could significantly contribute to the overall WSA health impacts.

The ER activity assay results demonstrate the presence of relatively low levels of endocrine active chemicals (EACs) or low potency EACs in WSA to which inhalation exposure could occur. WSA exposure risk would need to take into account a variety of aspects, including the ambient dose and the physicochemical characteristics and metabolic stability of the responsible EACs.

In terms of ecological risk, the concentration of WSA chemicals from partitioning into water would be expected to be low, and thus it would be unlikely to produce endocrine effects in wildlife from this route. In terms of exposure risk to humans, the comparison is challenging because the concentrations of known EACs are not as widely measured in $PM_{2.5}$ as the "usual suspects" like PAHs, and many EACs still remain to be identified. Additionally, the ability of a chemical to act as an EAC in vitro does not address whether it can produce adverse endocrine-related health effects in vivo (i.e. whether it acts as an

endocrine disruptor chemical (EDC)). In an indoor environment the WSA concentration can be higher than outdoors, but the exposure from using cookstoves is not as sustained over many hours or days such as that which occurs with a wildfire, and is not nearly as continual as with exposure to ER-active chemicals from consumer or industrial products. It's possible that risks from WSA exposure may be comparable to other EACs/EDCs of concern in some situations. It has been reported that $PM_{2.5}$ exposure can affect sperm development in humans (Wu et al., 2017; Lao et al., 2018), particularly in men who were exposed

to wintertime air pollution where WSA is high (Selevan et al., 2000), however, whether ER- and/or AhR-active chemicals present in smoldering WSA could contribute to this endocrine effects remains an open question. The relatively high AhR-dependent activities and relatively low ER-dependent activities detected in smoldering WSA are similar to previous observations of ambient air and aerosols from polluted sites (Novák et al., 2009; Novák et al., 2014).

The relative biological effects of WSA from different woods is a novel insight from this study to the best of our knowledge.

From **Table 1**, it appears that the WSA extracts that are highly active in the AhR bioassay do not necessarily kill cells, and vice versa. In fact, we found that the cell toxicity and AhR activity of woodsmoke WSA were negatively correlated with a



fairly strong adjusted $R^2$ coefficient of 0.74 (**Fig. S5a**). This suggests that the smoldering WSA analyzed in these bioassays contain compounds whose cytotoxic activity is not AhR mediated, but may be classified among the myriad other AhR active chemicals that have been identified (Denison and Nagy, 2003; Galati and O'brien, 2004; Nguyen and Bradfield, 2007; DeGroot

et al., 2011). We found that the biological effects of WSA have a moderate ($R^2 > 0.5$) relationship with the percent by mass of the wood fuel that has been burned in a set duration of time (**Fig. 2**), which may be related to the lignin content and solvent-soluble monomers in the fuel (White, 2007). As all WSA effects have been normalized by mass, this indicates that the biological properties may be related to the degree of oxidation/degradation of the woods during the smoldering process. That said, we found no relationship between aerosol properties and the O:C ratio of the WSA compounds. This may be because

even oxidative processes can fragment molecules in the wood in addition to functionalizing certain molecules with oxygenated moieties. This interesting relationship should be further explored with more focused studies.

The MAC values for different WSA in this work are within range of organic-rich brown carbon (Updyke et al., 2012) and particulate matter from a smoldering fire (Patterson and McMahon, 1984), but lower than ambient brown carbon likely because ~ 90% of that absorption comes from elemental carbon (Washenfelder et al., 2015). The MAC for the extracts in our study

may also be lower than those extracted in less-polar solvents such as acetone (Chen and Bond, 2010). Within the limited fuel types employed in this work, there does not appear to be an obvious correlation between $MAC_{vis}$ and wood hardness or percent of wood burned by weight ($R^2 \sim 0.08$) – suggesting visible light absorption of the aerosols may depend on the more specific chemistry to form chromophoric constituents in the combustion. Furthermore, the $MAC_{vis}$ values do not correlate to either cell toxicity (**Fig. S5b**, adj. $R^2$ 0.03) or AhR activity (**Fig. S5c**, adj. $R^2$ -0.06), suggesting that chromophoric species (e.g.,

specific conjugated compounds, charge transfer complexes, etc.) are not necessarily those producing these effects.

### 3.2. Molecular composition of WSA and correlations to aerosol properties

The chemical composition of aerosols from the oxygen-poor smoldering combustion or oxygen-free pyrolysis of lignocellulose material such as wood has been the subject of extensive study (Edye and Richards, 1991; Simoneit et al., 1993; McKenzie et al., 1994; Ingemarsson et al., 1998; Simoneit et al., 2000; Oros and Simoneit, 2001a, b; Simoneit, 2002; Hosoya et al., 2007b;

Nunes et al., 2010), with many of the previous works focusing on the mid- to low-polarity constituents that can be analyzed by GC-MS. Low-intensity combustion abundantly forms phenolic derivatives with hydroxyphenyl (H), guaiacyl (G), and syringyl (S) units from the breakdown of lignin. Furan derivatives (e.g., furfurals), sugar anhydrides (e.g., levoglucosan, mannosan, galactosan), and other products are formed from the combustion of cellulose under a variety of conditions (Shafizadeh and Fu, 1973; Simoneit et al., 1999). PAHs were not expected to be formed at the 330 °C combustion temperature

in this work (Rhee and Bratzler, 1968; Sharma and Hajaligol, 2003), and indeed the GC-MS analysis using ethyl acetate solvent confirmed that PAHs were not observed above the detection limit (**Table S1**). If formed, PAHs might have ended up in tar-like material as opposed to the aerosols that were collected and tested. Thus, the biological activity and light absorption of the smoldering WSA in this work are not likely due to PAHs. However, we observed molecular formulas that could be assigned





to PAH-like compounds (**Fig. 4**) from lignin decomposition. The GC-MS analysis identified sugar anhydrides, phenolic
compounds, and alkane derivatives in the mid- to low-polarity fraction of the smoldering WSA (**Fig. S6**, **Table S1**), consistent
with previous reports.

**Figure 3** shows the positive ion mode HRMS spectra of WSA extracts, highlighting the more abundant constituents in the
polar, lower-volatility fraction. $MS^n$ analyses support that the majority of smoldering WSA compounds are phenolic species
(e.g., $CH_3OH$ losses, $H_2O$ losses, and phenyl ionic fragments) that have a variety of carbonyl, alcohol, alkenyl, acid, and other
moieties. There are 300 – 400 peaks in each spectrum that are both over the limit of detection and that can be assigned to a
molecular formula within 2 ppm mass accuracy. There are more than 700 unique peaks observed in all samples.

**Figure 4** shows the chemical structures of select observed compounds. The compound with the highest abundance in most
samples was sinapaldehyde ($C_{11}H_{12}O_4$, **Fig. 3** green), except for the coniferous softwoods (Pine/Redwood) where
coniferaldehyde ($C_{10}H_{10}O_3$, **Fig. 3** blue) was the highest peak. This is consistent with the fact that coniferyl alcohol is the main
polymer building block of softwood lignin, while sinapyl alcohol (to a higher extent) and coniferyl alcohol are both important
building blocks of hardwood lignin (Graglia et al., 2015). We did not observe high abundance of hydroxyphenyl (H)
derivatives, (e.g., coumaraldehyde is ~ 25 times less abundant than sinapaldehyde on average) which is consistent with the
fact that coumaryl alcohol is the dominant lignin building block in grasses instead of wood (Himmelsbach and Barton, 1980).
Levoglucosan (and its isomeric sugar anhydrides, $C_6H_{10}O_5$, **Fig. 3** magenta) was also abundantly observed in HRMS, with the
largest signal in Redwood WSA, alongside its decomposition products such as furfural, hydroxymethylfurfural,
tetrahydrofuran derivatives, and others (Hosoya et al., 2007a; Lin et al., 2009).

High-molecular weight compounds that were observed are tentatively assigned to dimers from phenolic G and S units building
blocks with various linkages (**Fig. 4,** bottom) according to previous observations (Goñi and Hedges, 1992; van der Hage et al.,
1994; Guillén and Ibargoitia, 1999; Christensen et al., 2017), mechanistic feasibility (Beste, 2014), and $MS^n$ evidence where
available. Radical chemistry of the lignin formation and combustion is probabilistic, such that the diversity of isomers increases
with molecular mass, so there are likely multiple structures possible for each larger molecular formula. For example, $C_{14}H_{14}O_4$
is assigned to 5,5'-diguaiacol based on its $C_7H_8O_2$ guaiacol neutral loss fragment but concurrent losses of CO and $C_2$-$C_3$
fragments indicate the presence of other species with the same elemental makeup. Many of the proposed dimer assignments
can be rationalized with linkages that are already present in the wood itself (e.g., β-O-4, β-β, β-5, 5-5', (Watts et al., 2011)),
suggesting that simple depolymerization of the lignin plays an important role in forming aerosol-phase species, similarly to
pyrolysis. Attempts to perform $MS^n$ on most high-molecular-weight peaks were inconclusive due to the higher density of
accurate-mass peaks at each m/z that provided a challenge for their isolation by the ion trap. Chemical assignments without
associated $MS^n$ study in this work should be treated as purely speculative. A representative $MS^2$ spectrum and proposed
fragmentation loss pathways are shown in **Figure S7**. The full list of common ions, regardless of signal or correlations, is
shown in **Table S2**.



We then performed linear least-squares correlations of the accurate-mass m/z peaks with the aerosol properties of cell toxicity, AhR activity, and visible light absorption ($MAC_{vis}$) for each WSA sample. A potential concern is that HRMS signals in complex mixtures can suffer from matrix effects so that the signal is not necessarily representative of concentration; however, we found the HRMS signals are strongly correlated with GC-MS signals (e.g., sinapaldehyde $[M+H]^+$ m/z = 209.081, GC

retention time 16.6 minutes, has adj. $R^2$ = 0.81, **Fig. S8**), suggesting that the matrix is similar enough between the different samples that correlations using HRMS are likely meaningful. Peaks are required to be present in 8 out of 9 samples for the statistical analysis. **Table 2** shows the correlation results, the m/z and neutral molecular formulas, neutral or radical loss fragments derived from CID, and proposed assignments for select peaks based on the $MS^2$ evidence when available, literature previously cited in this article, and guidance from GC-MS observations (**Table S1**).

The occurrence of structural isomers is a limitation that may confound the statistics in this work, especially if those isomeric compounds have opposing effects on bioactivity or light absorption. With those caveats in mind, we observed a number of moderate-to-strong correlations with exact m/z peaks, either with a positive or negative effect, for the three aerosol properties examined. A positive effect means that the signal of the peak directly correlates (higher signal = lower cell toxicity $EC_{50}$, higher AhR activity, higher $MAC_{vis}$) and a negative effect means anti-correlation, neither of which is necessarily causative.

For example, levoglucosan was found to be anti-correlated with $MAC_{vis}$ (**Fig. 5**, $R^2$ = 0.75) and we interpret this to mean that lignin derivatives are more likely responsible for visible light absorption because of their highly-conjugated structures versus cellulose derivatives (**Fig. 4**), so when a wood has more cellulose content (producing more levoglucosan), the $MAC_{vis}$ decreases. Also, for example, we found the m/z of vanillin to be somewhat correlated with AhR activity ($R^2$ = 0.45), and even though vanillin itself is not AhR active, Bartonkova and Dvorak (2018) found that the complex mixture of vanilla is highly

AhR active, postulating that the AhR activity of the vanilla mixture is caused by minor constituent(s) (< 10%). Similarly, our correlation of vanillin with AhR activity may be due to vanillin's interaction or co-formation with other constituents in the WSA extract.

The cell toxicity correlations appears to be mainly driven by $C_{11}$ compounds, some of which are syringyl derivatives. We found a positive effect of sinapaldehyde and sinapyl alcohol on cell toxicity, and possibly for related compounds such as $C_{11}H_{14}O_5$

that is tentatively assigned to be sinapyl hydroperoxide. The hydroperoxide assignment is based on the facile $H_2O$ loss (MacMillan and Murphy, 1995) with lack of HCOOH, and a feasible mechanism of formation, e.g., C-O cleavage of sinapyl alcohol in a heated oxidative environment to form the R radical (Beste, 2014) that can add $O_2$ and then form the ROOH through reaction with $HO_2$ or other RH (Atkinson, 2000). This is somewhat consistent with sinapaldehyde's strong effect on inhibiting the growth of bacterial cells, while other phenolic compounds like vanillin and syringaldehyde had no effect (Figueiredo et al.,

2008). Similarly syringaldehyde and vanillin had no effect in our cell toxicity assay. The potential toxicity of monolignols like sinapyl alcohol has been noted in plants (Whetten and Sederoff, 1995), and while we would expect that coniferyl alcohol may have a similar effect, no correlation was observed. Among other factors, this could be due to isomeric compounds present at


the coniferyl alcohol m/z, including an acid (HCOOH loss) and aldehyde (CO loss), compared to the sinapyl alcohol peak that did not have these interferences. Another positively correlated molecular formula is $C_{14}H_{16}O_2$ ($R^2$ = 0.74) that appears to belong to compounds with quinone, furfuryl alcohol, and other substructures in the WSA based on fragmentation patterns so it is not clear which compounds specifically effect cell toxicity. Quinones can be cytotoxic or cytoprotective depending on structure (Bolton and Dunlap, 2016) and furfuryl alcohol can be mildly immunotoxic (Franko et al., 2011); therefore, the correlation has mechanistic plausibility. We found that 5-hydroxymethylfurfural (HMF), an important decomposition product of cellulose, was anticorrelated with cell toxicity ($R^2$ = 0.54) for reasons that are unclear. HMF can be biologically active; however, at a dose that would be unfeasibly high for human exposure (~ 200 g based on the oral acute $LD_{50}$ of 2.5 – 3.1 g/kg in rats (Hoydonckx et al., 2000)) and we would expect it to have no effect on cell toxicity in our assay. It is possible that HMF is anti-correlated because of its connection with levoglucosan, which is also anti-correlated but with a weaker relationship ($R^2$ = 0.31) and consequently not included in **Table 2.** If so, the interpretation would be similar to that for $MAC_{vis}$, i.e., a higher lignin/cellulose ratio in the wood may increase cell toxicity in the smoldering WSA.

In this study, compounds that correlate with AhR activity tend to have more carbons than the lignin monomers (mainly $C_{14}$-$C_{19}$) and more C=C unsaturation. These compounds are either formed in the combustion or exist naturally in wood, as wood dust itself has been shown to be AhR-active (Wilson et al., 2015). The calculated double bond equivalents (rings + π-bonds = DBE, (Badertscher et al., 2001)) are consistent with AhR active compounds having more unsaturation on average. DBEs for the positively-correlated compounds (<DBE> = 8.0, max 11) are notably higher than those that are not correlated (<DBE> = 6.2, max 11) and those that are anti-correlated (<DBE> = 5.6, max 7). This explanation is an oversimplification, as double bonds and even aromaticity do not account for the specific electronic properties or the conformational planarity of the ligands that are key to effective binding with AhR (Denison et al., 2002; Petkov et al., 2010). If the $C_{14}$-$C_{19}$ compounds are indeed lignin dimers, it is reasonable to postulate that the higher AhR activity is partially due to the ability of the dimers to form PAH-like structures from the radical reactions. For example, $C_{16}H_{14}O_4$ (**Fig 4** bottom right) is assigned to be (at least partially) a phenanthrene derivative generated from β-5 dimerization or radical depolymerization reactions and ring closure (Beste, 2014) and may be potent for induction of AhR – we found a fairly high correlation coefficient ($R^2$ = 0.65, **Fig. 5**). Similarly, other molecular formulas that can be assigned to a phenanthrene derivatives ($C_{18}H_{16}O_5$ and $C_{18}H_{16}O_6$) are also positively correlated with AhR activity (both $R^2$ = 0.4).

$C_{14}H_{12}O_3$ ($R^2$ = 0.57, positive effect for AhR and $R^2$ = 0.34, negative affect on cell toxicity (not shown on Table 2)) may be related to the naturally-occurring phenol resveratrol which is a dimer of H and dihydroxybenzene with a β-1 linkage. Resveratrol, found in wood in low amounts 0.017 – 0.271 g/kg, (Tyśkiewicz et al., 2019)) and other plant materials, has AhR antagonist activity and may be protective against dioxin (AhR-dependent) toxicity (Casper et al., 1999), consistent with the negative correlation observed here. The lower signal of this ion in our samples prevented a more-detailed $MS^n$ study, so the assignment is tentative. However, since this molecule has only alcohol groups, the $H_2O$ loss we observed in absence of $CH_3OH$,



CO, or other labile groups does not contradict the assignment. Some smaller compounds were also positively correlated with AhR activity, with the highest correlation for $C_{10}H_{10}O_2$ ($R^2 = 0.74$), assigned to methoxy cinnamaldehyde based on its relatively-clean $MS^2$ spectrum. The interaction of methoxy cinnamaldehyde with the AhR-responsive gene product CYP1A1 results in its oxidation into an acid (Hasegawa et al., 2002), and it appears to be mainly protective (Cope, 2019). This would be consistent with the observed anti-correlation of this product with cell toxicity ($R^2 = 0.42$).

Roughly 90% of the correlations with $MAC_{vis}$ are positive in nature and the positively-correlated compounds are fairly diverse, but many have conjugated structures. The conjugation that is observed for lignin phenols (DBE ~ 5) and even for the observed PAH-like compounds are insufficient to absorb visible light, however, when relatively-small lignin phenols are deprotonated in solution, they absorb light in the visible range due to the resonance stabilization of the phenolates – the red shift is much more pronounced for phenols with extra conjugation such as sinapaldehyde (Panossian et al., 2001). It is not clear why
sinapaldehyde and coniferaldehyde do not correlate with $MAC_{vis}$, but vanillydene acetone with its additional sigma carbon does. It could be that our solution is not sufficiently basic for the deprotonated form to substantially exist, but vanillydene acetone can achieve additional resonance with its keto-enol tautomerization (which tends not to happen in the ring because the aromatic phenol is highly favored). Even so, many of the correlations are not easily understood and could be due to coincidence or driven by unassigned isomers. In the ambient, nitrophenols may significantly contribute to wood-burning brown carbon
(Mohr et al., 2013). In this work, these nitrophenols were not observed by GC-MS, so the mechanism of light-absorption is likely different. It is possible that charge-transfer complexes between neighboring OH and C=O in dimers (e.g., $C_{18}H_{18}O_6$, **Fig. 4**) may be chromophoric, as suggested previously for brown carbon aerosols (Phillips and Smith, 2014). The smaller phenols may also potentially form intermolecular complexes with proton acceptors such as carbonyls or heterocyclic nitrogen species (e.g., $C_9H_{11}NO_4$ with a pyridine-like ($C_5H_5NO$) loss fragment), as the aerosols species are in close proximity. While no four-
ring PAH-like species that would absorb in the visible range (Samburova et al., 2016) were observed in this work, we can't rule out the possibility that highly-conjugated compounds below the detection limit can contribute to light absorption in combination with other chromophores. For the compounds that are correlated with $MAC_{vis}$, it is likely that they need to be activated to the ionic form to be chromophoric.

    Finally, although composition characterization was not the main focus of this work, we observed some fairly abundant
compounds that have not been previously reported. For example, we identified some organic nitrogen compounds as unknown pyridine derivatives (e.g., $C_9H_{11}NO_3$) and nitro or nitrate derivatives (e.g., $C_5H_{10}N_2O_8$), although in general, organic nitrogen is a larger fraction of ambient biomass burning aerosols (Laskin et al., 2009; Lin et al., 2016) than in this study. Poplar WSA also has high abundance peaks that have not been previously observed, e.g., corresponding to the molecular formula $C_{20}H_{21}NO_3$ (**Fig. 3**, dark yellow). Due to the high signal, we assigned $C_{20}H_{21}NO_3$ to one or more naturally-occurring alkaloids as opposed
to being formed from dimerization processes of phenols (plausible monomers should have higher signals than dimers). However, alkaloids that have been extracted from poplar heartwood with the same molecular formula (Chen-Loung et al.,



1976; Cordell et al., 1989) do not appear to be entirely consistent with the $MS^2$ fragmentation patterns of $C_{20}H_{21}NO_3$ in this work (neutral losses $CH_3OH$, $C_3H_6O_2$, and $C_3H_9NO_3$), suggesting that this compound may be different than previously-identified alkaloids or modified from an oxidative process in the smoldering combustion. Poplar WSA also have a relatively

low $EC_{50}$ value in our cell toxicity study, which is consistent with the higher cell toxicity and genotoxicity response of combustion aerosols from Poplar compared to softwood pellets in residential boilers (Kasurinen et al., 2016). Unfortunately, since other types of woods were not studied by Kasurinen and colleagues (2016), a more thorough comparison can not be made. As the HRMS correlations require the peaks to be present in most samples for adequate statistics, it is not clear whether the properties of the Poplar or other WSA are associated with the uniqueness of their composition rather than commonalities.

**4. Conclusions**

The composition of smoldering WSA generated in the conditions of this work produced abundant lignin and cellulose oxidation and decomposition products that impact aerosol properties in ways that are not easily predictable. We showed that the smoldering components of WSA have high biological activity that can substantially contribute to the environmental health burden of woodsmoke, and that these health-related effects are not likely PAH-mediated. The observed bioactivity may be

linked to the percent of the fuel that has been burned, and thus to the degree of combustion, with more toxic aerosols formed at earlier stages of the burn. It is possible that the more toxic compounds are eventually degraded during the combustion process. These results underscore the importance of untargeted analyses to move beyond well-studied toxicants when considering organic aerosol properties, as the statistical studies identified multiple targets for further toxicological testing. We found that lignin phenols are correlated with visible light absorption (i.e., levoglucosan is anti-correlated) and hypothesize that

the mechanism of action is through charge transfer reactions to form phenolates. The lignin percent of wood may also drive toxicity effects, but this remains to be explored. Although some mechanistically-probable correlations with bioactivity and light absorption are found in this work, the potential correlations are possibly obscured by isomeric interference at some masses. Future work should add a separation component to the accurate-mass and $MS^n$ analysis of WSA for additional clarity. This information will help understand the contributions of individual components and enhance the value of testing such

complex mixtures.

**5. Data Availability**

Data are available from the corresponding author upon request.

**6. Competing interests**

The authors declare no competing financial interests.




## 7. Author Contributions

TBN and KQN designed the experiments, LKC and KQN carried out the experiments. All authors contributed original data and data analyses. LKC, KQN, and TBN prepared the draft manuscript. All co-authors have reviewed and edited the manuscript.

## 8. Acknowledgements

This work was partially supported by the National Science Foundation grant AGS1656889; a Superfund Research Grant from the National Institute of Environmental Health Sciences (P42ES004699).

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





**Table 1:** *Results for cellular assays and visible light mass absorption coefficient (MAC$_{vis}$) for WSA extracts. Lower EC$_{50}$ values correspond to higher toxicity. AhR and ER activities are expressed as bioequivalent (BEQ) values of calibrant compounds 2,3,7,8-tetrachlorodibenzo-p-dioxin (TCDD) and 17β-estradiol (E2), respectively. All aerosol properties reference the mass in grams of original WSA material that were tested. The concentration-response curves for the cell toxicity of different WSA (**Fig. S3**) and receptor activity of TCDD and ER (**Fig. S4**) can be found in the supplement.*

| Wood sample | Cell Toxicity EC$_{50}$ (μg) | AhR Activity (TCDD BEQ, g/g) | ER Activity (E2 BEQ, g/g) | MAC$_{vis}$ (cm$^2$/g) |
|---|---|---|---|---|
| Afr. Mahogany | $250 \pm 20\%$ [a] | $2.5 \times 10^{-6} \pm 12\%$ | $3.5 \times 10^{-9} \pm 15\%$ | $2009 \pm 10\%$ |
| Birch | $160 \pm 25\%$ | $1.9 \times 10^{-7} \pm 13\%$ | N.D. [b] | $1645 \pm 10\%$ |
| Cherry | $208 \pm 30\%$ | $2.2 \times 10^{-7} \pm 12\%$ | $3.5 \times 10^{-9} \pm 17\%$ | $1387 \pm 10\%$ |
| Maple | $160 \pm 20\%$ | $1.3 \times 10^{-6} \pm 10\%$ | $3.5 \times 10^{-9} \pm 10\%$ | $1927 \pm 10\%$ |
| Pine | $256 \pm 20\%$ | $4.0 \times 10^{-6} \pm 25\%$ | $1.8 \times 10^{-9} \pm 26\%$ | $1477 \pm 10\%$ |
| Poplar | $163 \pm 40\%$ | $7.5 \times 10^{-7} \pm 14\%$ | $1.5 \times 10^{-9} \pm 35\%$ | $1557 \pm 10\%$ |
| Red Oak | $109 \pm 40\%$ | $8.1 \times 10^{-8} \pm 16\%$ | N.D. | $1094 \pm 10\%$ |
| Redwood | $320 \pm 20\%$ | $3.4 \times 10^{-6} \pm 10\%$ | $3.6 \times 10^{-9} \pm 25\%$ | $637 \pm 10\%$ |
| Walnut | $320 \pm 20\%$ | $4.0 \times 10^{-6} \pm 25\%$ | $1.9 \times 10^{-9} \pm 30\%$ | $1118 \pm 10\%$ |

a. *Table values are reported as the mean of repeated trials with uncertainty noted as percent of the mean that has been propagated from multiple sources*

b. *Not detected (N.D.). Values are not significantly different than controls at this sample preparation protocol.*





**Table 2:** *Peaks observed in at least 8 out of 9 samples with either correlations to aerosol properties or other significance (e.g., high signals). The average normalized signal-to-noise in all samples and the neutral molecular formulas are shown.*

| Obs. m/z | S/N (%) | Formula C | H | O | N | Tox Sign | R² | AhR Sign | R² | MAC Sign | R² | Proposed chemical assignment(s) | CID neutral or radical losses |
|---|---|---|---|---|---|---|---|---|---|---|---|---|---|
| 127.039 | 1.3 | 6 | 6 | 3 | 0 | neg | 0.54 | pos | 0.40 | --- | --- | 5-hydroxymethyl furfural | $H_2O$, CO, $C_2H_2O_2$ ($CH_2O$ + CO?) |
| 129.054 | 0.3 | 6 | 8 | 3 | 0 | neg | 0.58 | --- | --- | --- | --- | --- | --- |
| 137.059 | 1.3 | 8 | 8 | 2 | 0 | --- | --- | --- | --- | --- | --- | toluic acid, anisaldehyde | $CH_3$, $H_2O$, CO, HCOOH, $CH_2O$+CO |
| 139.075 | 0.2 | 8 | 10 | 2 | 0 | --- | --- | --- | --- | --- | --- | methylguaiacol | |
| 141.054 | 0.3 | 7 | 8 | 3 | 0 | --- | --- | --- | --- | pos | 0.52 | methoxymethyl-furfural, others | CO, $CH_4O$, $CH_2O$ + CO |
| 143.034 | 0.8 | 6 | 6 | 4 | 0 | --- | --- | --- | --- | --- | --- | 5-Hydroxymethyl-2-furancarboxylic acid | HCOOH, $H_2O$ |
| 143.070 | 0.2 | 7 | 10 | 3 | 0 | --- | --- | --- | --- | pos | 0.46 | methoxyfuranethanol, hydroxymethyfuranethanol | $CH_3OH$, H2O |
| 149.059 | 0.2 | 9 | 8 | 2 | 0 | --- | --- | --- | --- | --- | --- | coumaraldehyde | --- |
| 151.039 | 0.6 | 8 | 6 | 3 | 0 | neg | 0.46 | pos | 0.44 | neg | 0.47 | hydroxyphenylglyoxal | $C_2H_2O$, CO, $H_2O$ |
| 151.075 | 0.7 | 9 | 10 | 2 | 0 | --- | --- | --- | --- | --- | --- | methoxyvinylphenol, coumaryl alcohol | CO, $CH_3OH$, $H_2O$, $C_3H_6$, $C_3H_4O$, $C_2H_4O_2$ |
| 153.054 | 0.8 | 8 | 8 | 3 | 0 | neg | 0.47 | pos | 0.45 | --- | --- | vanillin (minor methoxybenzoic acid) | CO, $CH_3OH$ (minor HCOOH, $CH_3OH$+$CO_2$) |
| 153.091 | 0.2 | 9 | 12 | 2 | 0 | --- | --- | --- | --- | pos | 0.59 | hydroxypropylphenol | $H_2O$, $C_3H_8O$ |
| 155.070 | 0.6 | 8 | 10 | 3 | 0 | --- | --- | --- | --- | --- | --- | syringol, hydroxyl-ethylbenzene diol | $CH_3OH$, $CH_3$, $H_2O$, $C_2H_4O_2$ |
| 163.039 | 0.7 | 9 | 6 | 3 | 0 | --- | --- | --- | --- | --- | --- | hydroxycoumarin | CO, $CO_2$ |
| 163.075 | 0.7 | 10 | 10 | 2 | 0 | neg | 0.43 | pos | 0.72 | --- | --- | methoxy-cinnamaldehyde | CO, $CH_3OH$, $C_2H_2O$, $C_3H_4O$ |
| 165.091 | 0.4 | 10 | 12 | 2 | 0 | --- | --- | --- | --- | --- | --- | eugenol, isoeugenol | $H_2O$, $CH_3OH$, $C_3H_6$ |
| 167.070 | 2.7 | 9 | 10 | 3 | 0 | --- | --- | neg | 0.44 | --- | --- | homovanillin, acetovanillone | $CH_3$, $H_2O$, CO, $CH_3OH$, $C_2H_2O$, $CH_2O_2$, $C_2H_4O_2$ |
| 169.049 | 0.4 | 8 | 8 | 4 | 0 | --- | --- | --- | --- | --- | --- | vanillic acid, dihydroxyphenylacetic acid | CO, $H_2O$, $CH_3$, HCOOH, $C_2H_2O$,$C_2H_4O_2$ |
| 169.086 | 0.6 | 9 | 12 | 3 | 0 | --- | --- | --- | --- | pos | 0.46 | homovanillyl alcohol | $C_2H_4O$, $H_2O$, $CH_3OH$ |
| 177.054 | 1.2 | 10 | 8 | 3 | 0 | --- | --- | --- | --- | --- | --- | methoxycoumarin, hydroxymethyl-coumarin | CO, $CH_3OH$, $C_4H_4O$, $CH_3$ |
| 177.091 | 0.3 | 11 | 12 | 2 | 0 | --- | --- | --- | --- | pos | 0.59 | ethylcinnamate | $H_2O$, $C_2H_4$, $C_2H_2O$ |
| 179.070 | 3.0 | 10 | 10 | 3 | 0 | --- | --- | --- | --- | --- | --- | coniferaldehyde, methoxycinammic acid | $H_2O$, CO, $CH_3OH$, HCOOH, $C_2H_2O$, $C_3H_4O$ |
| 179.106 | 0.1 | 11 | 14 | 2 | 0 | --- | --- | --- | --- | pos | 0.77 | --- | CO, $C_2H_4O$, $C_2H_4O_2$ (CO + $CH_3OH$?) |



| m/z | % | C | H | O | N | m1 | v1 | m2 | v2 | m3 | v3 | Identification | Formulas |
|---|---|---|---|---|---|---|---|---|---|---|---|---|---|
| 181.086 | 1.3 | 10 | 12 | 3 | 0 | --- | --- | --- | --- | pos | 0.51 | coniferyl alcohol, dihydroconiferald-ehyde, methoxyphenyl-propanoic acid | $H_2O$, $CH_3$, $CO$, $C_2H_4$, $C_2H_2O$, $C_3H_6$, $HCOOH$, $C_2H_6O$, $C_3H_4O$, $C_2H_4O_2$, $C_3H_6O_2$ |
| 183.065 | 1.8 | 9 | 10 | 4 | 0 | --- | --- | --- | --- | --- | --- | syringaldehyde, homovanillic acid | $H_2O$, $CO$, $HCOOH$, $CH_3OH$, $C_2H_4O_2$ |
| 183.101 | 0.4 | 10 | 14 | 3 | 0 | --- | --- | --- | --- | pos | 0.76 | dihydroconiferyl alcohol, others | --- |
| 185.042 | 1.2 | 6 | 10 | 5 | 0 | --- | --- | --- | --- | neg | 0.75 | levoglucosan, mannosan, galactosan | $H_2O$ |
| 187.075 | 0.2 | 12 | 10 | 2 | 0 | --- | --- | --- | --- | pos | 0.47 | phenyl furancarbaldehyde | $CO$,$C_5H_5O_2$ |
| 191.070 | 1.2 | 11 | 10 | 3 | 0 | --- | --- | --- | --- | pos | 0.64 | acetyl coumaraldehyde | $CO$, $H_2O$, $CH_3$, $C_2H_4O_2$ |
| 191.106 | 0.2 | 12 | 14 | 2 | 0 | --- | --- | --- | --- | pos | 0.49 | isoeugenyl acetone | --- |
| 193.049 | 0.6 | 10 | 8 | 4 | 0 | --- | --- | --- | --- | neg | 0.47 | hydroxy(oxopropenyl)benzoic acid | $CH_3$, $CO$, $H_2O$, $HCOOH$ |
| 193.086 | 1.4 | 11 | 12 | 3 | 0 | --- | --- | --- | --- | pos | 0.78 | vanillylidene acetone, dimethoxyphenylacrylaldehyde | $CH_3OH$, $H_2O$, $CO$, $C_2H_2O_2$, $C_3H_4O$, $C_2H_4O_2$, $C_2H_4O$, $C_3H_4O_3$ |
| 195.101 | 0.7 | 11 | 14 | 3 | 0 | pos | 0.55 | neg | 0.41 | pos | 0.57 | methoxyisoeugenol, vanillylacetone | --- |
| 197.080 | 1.7 | 10 | 12 | 4 | 0 | --- | --- | --- | --- | --- | --- | acetosyringone | $C_2H_2O$, $H_2O$, $CH_3OH$, $C_2H_2O+H_2O$ |
| 197.117 | 0.1 | 11 | 16 | 3 | 0 | --- | --- | --- | --- | pos | 0.80 | --- | --- |
| 198.076 | 0.7 | 9 | 11 | 4 | 1 | --- | --- | --- | --- | --- | --- | pyridine derivative | $C_4H_6O_3$, $H_2O$, $C_5H_5NO$ |
| 199.060 | 0.2 | 9 | 10 | 5 | 0 | --- | --- | --- | --- | --- | --- | syringic acid | --- |
| 201.091 | 0.2 | 13 | 12 | 2 | 0 | --- | --- | --- | --- | pos | 0.63 | --- | --- |
| 205.084 | 1.1 | 10 | 14 | 3 | 0 | --- | --- | --- | --- | --- | --- | dihydroconiferyl alcohol, unknown aldehyde | $CH_3$, $H_2O$, $CO$, $CH_3OH$, $C_2H_2O$, $HCOOH$, $C_3H_4O$, $C_2H_2O_2$ |
| 207.065 | 0.7 | 11 | 10 | 4 | 0 | --- | --- | --- | --- | --- | --- | --- | $H_2O$ |
| 207.101 | 0.6 | 12 | 14 | 3 | 0 | --- | --- | --- | --- | pos | 0.70 | acetyleugenol | $CH_3OH$, $C_2H_4O_2$ |
| 209.080 | 5.8 | 11 | 12 | 4 | 0 | pos | 0.49 | neg | 0.51 | --- | --- | sinapaldehyde (minor dimethoxycinammic acid) | $H_2O$, $CO$, $CH_3$, $CH_3OH$, $C_2H_2O$, $C_3H_4O$, $HCOOH$, $CH_2O+CH_2O$ |
| 209.117 | 0.1 | 12 | 16 | 3 | 0 | --- | --- | --- | --- | pos | 0.88 |  | --- |
| 211.096 | 3.7 | 11 | 14 | 4 | 0 | pos | 0.44 | neg | 0.56 | --- | --- | sinapyl alcohol | $H_2O$, $CH_3OH$, $CH_2O+CH_2O$, $C_3H_6O_2$ (maybe $C_3H_4O + H_2O$) |
| 219.101 | 0.8 | 13 | 14 | 3 | 0 | --- | --- | pos | 0.41 | --- | --- | unknown phenols, G-methylfuran (α-2) dimer | $CH_3$, $H_2O$, $CO$, $CH_3OH$, $C_2H_2O$, $HCOOH$, $C_3H_4O$, $C_2H_2O_2$, $C_3H_6O_2$, $C_5H_6O$, $C_3H_4O_3$, $C_7H_8O_2$ |
| 221.080 | 0.5 | 12 | 12 | 4 | 0 | --- | --- | --- | --- | --- | --- | acetoconiferaldehyde | $CH_3$, $H_2O$, $CO$, $CH_3OH$, $C_2H_2O$, $HCOOH$,$C_2H_4O_2$ |
| 221.117 | 0.2 | 13 | 16 | 3 | 0 | --- | --- | --- | --- | pos | 0.94 | eugenyl propionate, and another phenol | $H_2O$, $CO$, $CH_3OH$, $C_3H_6$, $C_3H_6O_2$ |



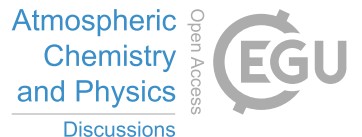

| m/z | | C | H | O | N | | | | | | | Identity | Neutral losses |
|---|---|---|---|---|---|---|---|---|---|---|---|---|---|
| 223.060 | 0.4 | 11 | 10 | 5 | 0 | pos | 0.49 | neg | 0.62 | --- | --- | --- | --- |
| 223.096 | 0.6 | 12 | 14 | 4 | 0 | pos | 0.54 | neg | 0.53 | --- | --- | tetrahydrofuran derivative | $C_4H_8O$, $C_5H_{10}O$ |
| 225.111 | 0.2 | 12 | 16 | 4 | 0 | pos | 0.57 | --- | --- | pos | 0.60 | --- | --- |
| 227.052 | 1.2 | 5 | 10 | 8 | 2 | neg | 0.43 | pos | 0.50 | neg | 0.58 | unknown nitro or nitrate derivative | $H_2O$, $CH_3OH$, $CH_5NO_2$ |
| 227.091 | 0.3 | 11 | 14 | 5 | 0 | pos | 0.69 | neg | 0.66 | --- | --- | sinapyl hydroperoxide | $H_2O$ |
| 229.084 | 0.2 | 14 | 12 | 3 | 0 | --- | --- | pos | 0.57 | --- | --- | resveratrol | $H_2O$ |
| 235.096 | 0.6 | 13 | 14 | 4 | 0 | --- | --- | --- | --- | pos | 0.56 | --- | --- |
| 237.111 | 0.3 | 13 | 16 | 4 | 0 | pos | 0.52 | neg | 0.44 | pos | 0.53 | --- | --- |
| 239.091 | 0.1 | 12 | 14 | 5 | 0 | pos | 0.52 | neg | 0.48 | --- | --- | --- | --- |
| 245.080 | 0.2 | 14 | 12 | 4 | 0 | neg | 0.50 | pos | 0.57 | --- | --- | --- | --- |
| 247.096 | 0.5 | 14 | 14 | 4 | 0 | --- | --- | --- | --- | --- | --- | diguaiacol (5,5'), others | $H_2O$, $CH_3$, $CH_3OH$, CO, $C_2H_4O_2$, $C_2H_2O$, HCOOH, $C_3H_4O_3$, $C_3H_4O$, $C_3H_6O_2$, $C_6H_6O_2$, $C_7H_8O_2$ |
| 247.132 | 0.2 | 15 | 18 | 3 | 0 | --- | --- | --- | --- | pos | 0.41 | --- | --- |
| 249.111 | 0.7 | 14 | 16 | 4 | 0 | pos | 0.74 | neg | 0.60 | pos | 0.49 | furfuryl alcohol deriv., hydroxybenzoquinone derivative ($C_6H_4O_3^+$ fragment ion), others | $H_2O$, $C_8H_{12}O$, CO, $CH_3OH$, $C_6H_8O_2$, $C_3H_6O_2$, $C_2H_4O_2$, $C_4H_6O_3$, $C_4H_6O_3$, $C_5H_6O_2$, $C_4H_8O_2$, $C_3H_6$ |
| 251.127 | 0.1 | 14 | 18 | 4 | 0 | --- | --- | --- | --- | pos | 0.64 | --- | --- |
| 261.112 | 0.5 | 15 | 16 | 4 | 0 | --- | --- | --- | --- | --- | --- | G-guaiacol dimer (α-2), others | --- |
| 265.106 | 0.3 | 14 | 16 | 5 | 0 | --- | --- | --- | --- | pos | 0.46 | S-furfurylalcohol dimer (α-2) | --- |
| 271.096 | 0.3 | 16 | 14 | 4 | 0 | neg | 0.63 | pos | 0.65 | --- | --- | G-guaicol (β-5, ring closure) | --- |
| 273.112 | 0.2 | 16 | 16 | 4 | 0 | --- | --- | --- | --- | --- | --- | diguaicol (β-1), G-guaicol (β-5) | |
| 289.106 | 0.6 | 16 | 16 | 5 | 0 | --- | --- | --- | --- | --- | --- | G-syringol dimer (β-5), others | --- |
| 301.107 | 0.3 | 17 | 16 | 5 | 0 | neg | 0.41 | pos | 0.55 | --- | --- | G-vanillin dimer (β-5) | --- |
| 303.122 | 0.3 | 17 | 18 | 5 | 0 | neg | 0.44 | pos | 0.58 | --- | --- | guaicyl-syringyl dimer (stilbene), others | --- |
| 305.138 | 0.3 | 17 | 20 | 5 | 0 | --- | --- | --- | --- | pos | 0.67 | G-guaiacol dimer (β-O-4) | --- |
| 313.107 | 0.2 | 18 | 16 | 5 | 0 | --- | --- | pos | 0.40 | --- | --- | G-homovanillin dimer (β-5, ring closure), others | --- |
| 315.122 | 0.3 | 18 | 18 | 5 | 0 | --- | --- | pos | 0.50 | --- | --- | G-homovanillin dimer (β-5), others | --- |
| 317.138 | 0.2 | 18 | 20 | 5 | 0 | --- | --- | pos | 0.43 | --- | --- | G-G dimer (α-5) | --- |
| 327.122 | 0.3 | 19 | 18 | 5 | 0 | --- | --- | --- | --- | --- | --- | G-coniferaldehyde dimer (β-1), others | --- |
| 329.102 | 0.1 | 18 | 16 | 6 | 0 | --- | --- | pos | 0.40 | --- | --- | S-vanillin dimer (β-5 ring closure), hydroxy G- | --- |





| | | | | | | | | | | | | homovanillin dimer (β-5, ring closure), others | |
|---|---|---|---|---|---|---|---|---|---|---|---|---|---|
| 329.138 | 0.3 | 19 | 20 | 5 | 0 | --- | --- | pos | 0.42 | --- | --- | eugenol-vanillone dimer (5,5'), others | --- |
| 331.117 | 1.0 | 18 | 18 | 6 | 0 | --- | --- | --- | --- | pos | 0.44 | G-vanillin dimer (5,5'), others | --- |
| 333.133 | 0.4 | 18 | 20 | 6 | 0 | --- | --- | --- | --- | --- | --- | S-S dimer (β-1), G-vanillin dimer (β-O-4) | --- |
| 345.132 | 0.2 | 19 | 20 | 6 | 0 | --- | --- | --- | --- | pos | 0.60 | coniferylalcohol-vanillone dimer (5,5'), others | --- |
| 357.130 | 1.3 | 18 | 22 | 6 | 0 | --- | --- | --- | --- | --- | --- | G-syringol dimer  or S-guaiacol dimer (β-O-4) | --- |
| 359.148 | 0.2 | 20 | 22 | 6 | 0 | --- | --- | --- | --- | pos | 0.77 | diconiferylalcohol (5,5', β-β, or β-), and others | --- |
| 383.146 | 0.2 | 20 | 24 | 6 | 0 | --- | --- | --- | --- | --- | --- | diconiferylalcohol (β-O-4) | --- |






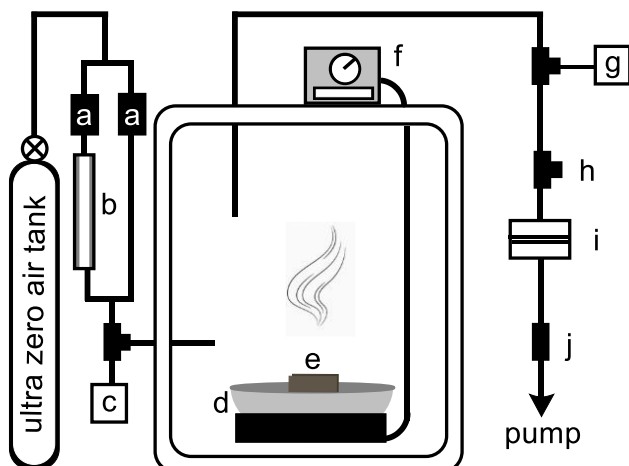

**Figure 1:** *Simplified diagram of smoldering combustion chamber. Key: (a) pressure flow restrictors, different flow rates each to control humidity; (b) humidifier tube; (c) relative-humidity probe; (d) heating mantle; (e) wood fuel blocks; (f) temperature controller; (g) thermocouple temperature measurement; (h) vent; (i) aerosol filter collection apparatus; (j) vacuum flow restrictor.*

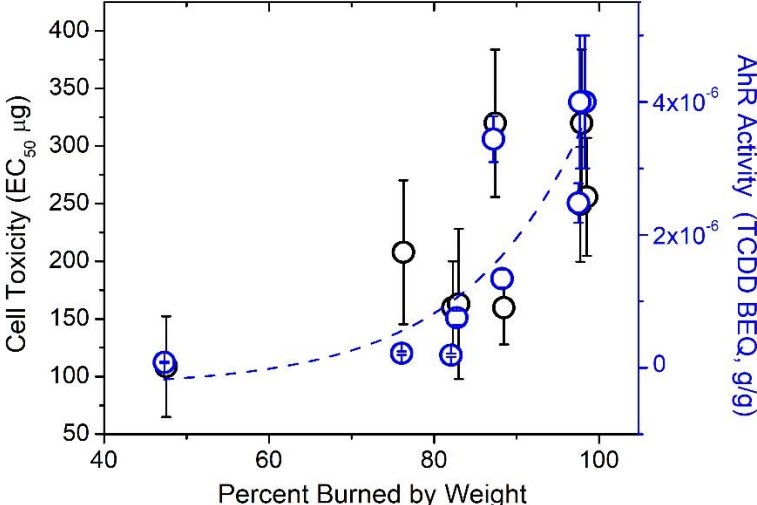

**Figure 2:** *Relationship between cell toxicity and AhR activity of smoldering WSA extracts with the percent of wood that has been burned at the time of aerosol collection.*



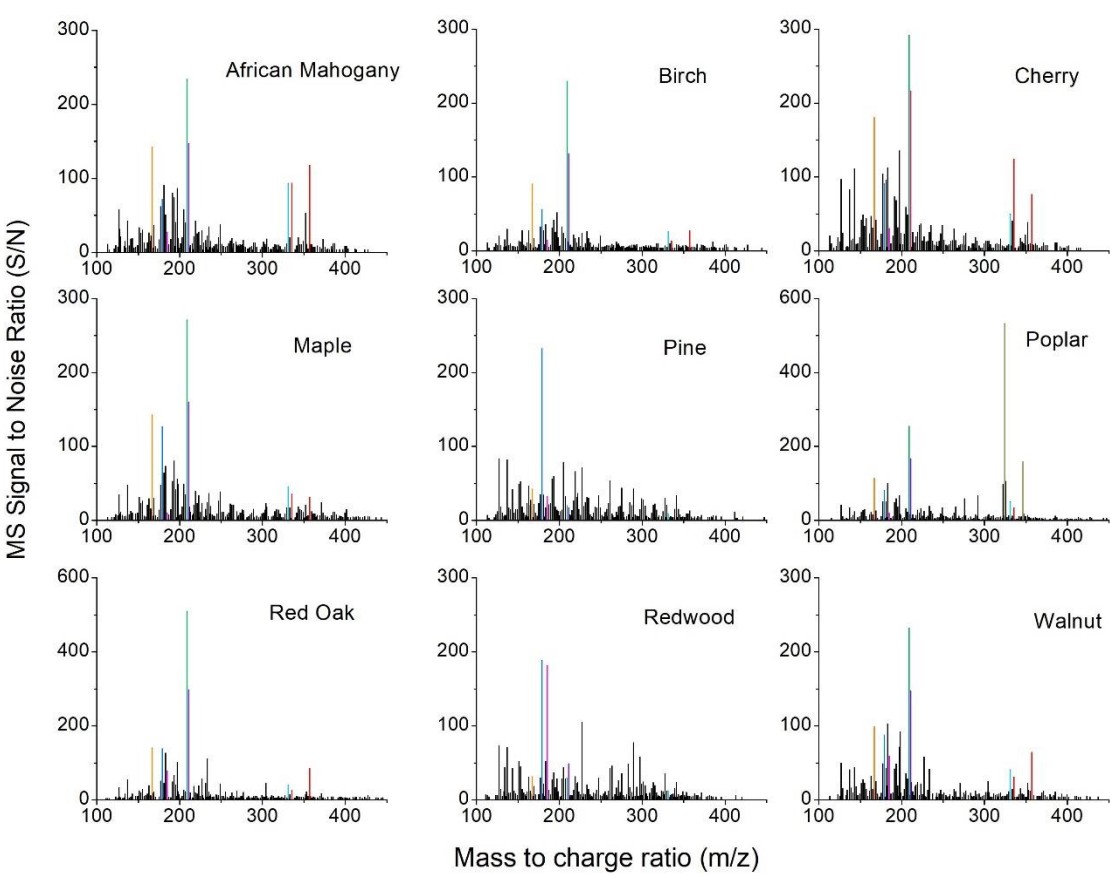

**Figure 3:** *High resolution mass spectra of WSA extracts in the study, taken at 100 ug/mL concentration. The color key for highlighted peaks with proposed assignments for the major species at each peak is: Orange – $C_9H_{10}O_3$, acetovanillone or homovanillin (protonated); Blue – $C_{10}H_{10}O_3$, coniferyl aldehyde (protonated); Magenta – $C_6H_{10}O_5$, levoglucosan, mannosan, or galactosan (sodiated); Green – $C_{11}H_{12}O_4$, sinapaldehyde (protonated); Cyan – $C_{18}H_{18}O_6$, G-vanillin dimer (protonated); Red – $C_{18}H_{22}O_6$, various S-G dimers (protonated and sodiated); Dark yellow – $C_{20}H_{21}NO_3$, unknown alkaloid (protonated and sodiated)..*







**Figure 4:** *Proposed chemical structures of some monomers and dimers observed in this work. More than one structure may be present at each molecular formula.*



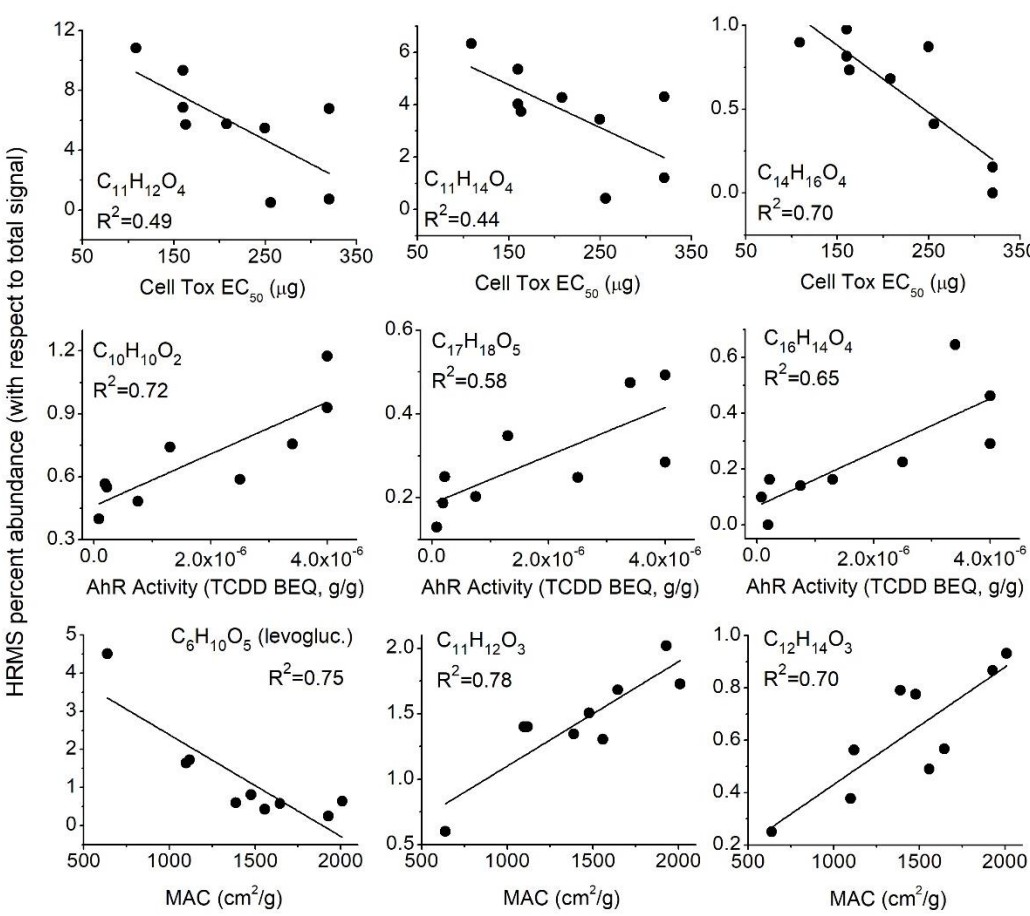

**Figure 5:** *Select correlations of HRMS peaks to cell toxicity EC$_{50}$, AhR activity, and MAC$_{vis}$. Linear fits and correlation coefficients are shown. Proposed chemical identities are listed in Table 1.*

770