# Peer review of "Relationship between the molecular composition, visible light absorption, and health-related properties of smoldering woodsmoke aerosols"

_Atmospheric Chemistry and Physics, 2019_

## Referee Comment (RC1) · Anonymous Referee #1 · 20 Oct 2019

This manuscript presents comprehensive molecular characterization of smoldering woodsmoke aerosols generated from controlled burning of different hardwood samples. Organic aerosol compositions were correlated with cell toxicity and visible light-absorption measurements to probe their potential health and climate effects. While this manuscript is overall well-written, there are several major concerns regarding the study design that need to be addressed before publication.

(1) The selection of cell type and biological endpoint for toxicity testing does not seem to be relevant to the major route of exposure for woodsmoke aerosol exposure. The authors stated that for the cell toxicity bioassay, human epidermal keratinocytes (i.e., skin

cells) were used to determine the toxicity of WSA, and the protein content from envelope formation was used as a quantitative biomarker from toxicant exposures. However, compared to dermal contact, inhalation should be more relevant. Use of pulmonary cell models to measure cytotoxicity (cell death or cell proliferation) would be more appropriate for this type of research. The authors should provide clear justifications for the usage of cell type and biological endpoint for toxicity testing.

(2) As discussed in section 3.1 (Line 188-201), the authors provided an exposure scenario, stating that a human adult would take $\sim$2175 breaths to reach the cell toxicity EC50 value of Red Oak smoldering woodsmoke aerosols. Again, since epidermal keratinocytes (i.e., skin cells) were used to determine cell toxicity by measuring the protein content from envelope formation, the EC50 values derived from this assay are not appropriate parameters for an exposure assessment through inhalation. The authors should limit the inhalation-related discussion based on their EC50 values.

(3) The toxicity bioassay was quantified colorimetrically at the wavelength of 495 nm. This wavelength overlaps with the visible brown carbon absorption. Did the author measure the absorption from aerosol sample extracts to account for the potential interference from brown carbon absorption at this wavelength?

(4) Line 158: What were the positive controls used in the cell toxicity bioassay to simulate the maximum activity?

(5) Line 245: The cell toxicity measured by the protein content from envelope formation was not a direct measure of cell death. How did the authors determine whether the WSA extracts kill cells?

(6) The rationale for measuring AhR and ER activity from woodsmoke aerosol exposure should be strengthen. What are the underlying hypotheses?

(7) Throughout the manuscript, a lot of discussions between molecular composition and biological endpoints (e.g., AhR activity) were based on correlations. The authors

should be careful about drawing conclusions from this relationship since correlation does not imply causation. Validation work should be carried out.

(8) For the brown carbon measurement, why did the author measure the visible light only (400-700 nm), but not the full range for tropospheric-relevant radiation (300-700 nm)? Note that most brown carbon constituents absorb light at near-UV wavelengths (300-400 nm) and to a lesser extent visible light.

(9) Line 380-385: What fractions of sinapaldehyde and coniferaldehyde converted to their enol forms in the extract solutions?

(10) Line 386: The authors could check on the pKa of these detected products and compare that to the pH of your solution.

(11) Line 423-425: Does the formation of phenolate occur only in the aqueous solution?

(12) Line 12 and 86: did the author mean to say "hardwood" here?

---

## Referee Comment (RC2) · Anonymous Referee #2 · 2 Nov 2019

This study investigated the molecular composition, optical and biological properties of woodsmoke aerosols. Correlations between some identified molecular formulas with MAC and bio endpoints were found. Overall, the analyses are sound, and the manuscript is well-written and easy to follow. I recommend that it can be published following some revisions.

1. I have the same concern as Referee#1 comments (1) and (2) for the selection of cell type. The author should illustrate the rationale to use human epidermal keratinocytes. Also, why such high passage number (25-35) were being used? Usually the cell passage number shouldn't be higher than 20, otherwise the cell response results can be

biased.

2. Line 149: cells were incubated with extracts of WSA for 48h. During the 48h incubation in water/acetonitrile solution, cells may not be alive although the author did not find envelope formation in negative control. Did the authors try other cell viability assays to test cell viability? Why the author used such a long incubation time?

3. Line 263: no correlation between chromophoric species with cell toxicity does not necessarily mean they don't contribute to produce cell toxicity, and vice versa. Indeed, there is a recent paper (Chen et al., es&t 2019) showing significant correlation between DTT activity and water-soluble brown carbon.

4. There are a lot of correlation analysis in the manuscript. The authors should also do statistical analysis to show how significant the correlations are.

5. The authors should do some comparisons of their cell toxicity, AhR activity and ER activity results with other previous studies using the same assays. Only looking at the numbers in table 1 cannot get a sense how toxic the WSA are. The authors shall compare their results with other types of aerosols or other toxicants.

Minor comment: 1. Figure S3, scale the x-axis to 0.1 to 4 mg/mL would be better to present the concentration-dependent curves.
* * *

---

## Author Comment (AC1) · 22 Nov 2019

We thank both referees for their thoughtful reviews of our work, which have significantly improved the clarity of the manuscript. Below, we address each referee comment and highlight specific changes made to the manuscript. Our author changes are in blue, and the enumerated referee comments are in black.

**Anonymous Referee #1**

This manuscript presents comprehensive molecular characterization of smoldering woodsmoke aerosols generated from controlled burning of different hardwood samples. Organic aerosol compositions were correlated with cell toxicity and visible light absorption measurements to probe their potential health and climate effects. While this manuscript is overall well-written, there are several major concerns regarding the study design that need to be addressed before publication.

(1)  The selection of cell type and biological endpoint for toxicity testing does not seem to be relevant to the major route of exposure for woodsmoke aerosol exposure. The authors stated that for the cell toxicity bioassay, human epidermal keratinocytes (i.e., skin cells) were used to determine the toxicity of WSA, and the protein content from envelope formation was used as a quantitative biomarker from toxicant exposures. However, compared to dermal contact, inhalation should be more relevant. Use of pulmonary cell models to measure cytotoxicity (cell death or cell proliferation) would be more appropriate for this type of research. The authors should provide clear justifications for the usage of cell type and biological endpoint for toxicity testing.

We appreciate the intent of this recommendation, which is to provide a better approximation to reality. Keratinocytes are found normally in the upper respiratory tract (oropharynx, laryngeal pharynx), throughout the oral cavity and are seen in the trachea as a result of insult (e.g., chronic smoke inhalation); thus, they are relevant to inhalation exposure. Furthermore, keratinocytes have been shown to be a useful model for responses common to epithelial cells (Rhim, 1989; Neugebauer et al., 1996; Rogers et al., 2001).

It should be kept in mind that the respiratory tract is populated by about 40 different cell types, all with different properties. Except for white blood cells, none can be serially cultured in their normal state. Cell lines are available derived from the respiratory tract, but these are generally neoplastic and considerably deviated from normal. The commonly used BEAS-2B cell line, for example, is aneuploid and was derived from human bronchial epithelial cells by transformation with an adenovirus-SV40 hybrid virus (Reddel et al., 1988).

The human SIK line that we use is minimally deviated from normal (one chromosomal aberration (Rice et al., 1993)). The response we are using exploits a property of keratinocytes that is part of their normal terminal differentiation program occurring as the cells become permeable to trypan blue dye (Green, 1977). Moreover, the lung is not the only target tissue of air pollutants. Pollutants are taken up into the blood stream and distributed throughout the body with various target cells including keratinocytes of the esophagus and even epidermis. No assay will successfully reproduce the *in vivo* environment, but the one we use is as good as or better than others. Please note that the

Ah receptor assay does not use cells from the respiratory tract but is an excellent and relevant assay nevertheless.

We have added the following justification to Section 2.4:

"*Keratinocytes can be found naturally in the upper respiratory tract, oral cavity, and formed in the trachea as a response to air pollution, cigarette smoke, and woodsmoke inhalation exposure (Plamenac et al., 1979b; Plamenac et al., 1979a; Lee et al., 1994; Moran-Mendoza et al., 2008). Moreover, they have been shown to be a useful model for responses common to epithelial cells (Rhim, 1989; Neugebauer et al., 1996; Rogers et al., 2001).*"

(2)   As discussed in section 3.1 (Line 188-201), the authors provided an exposure scenario, stating that a human adult would take ~2175 breaths to reach the cell toxicity EC50 value of Red Oak smoldering woodsmoke aerosols. Again, since epidermal keratinocytes (i.e., skin cells) were used to determine cell toxicity by measuring the protein content from envelope formation, the EC50 values derived from this assay are not appropriate parameters for an exposure assessment through inhalation. The authors should limit the inhalation-related discussion based on their EC50 values.

We have addressed this concern in (1) – it has been demonstrated in the literature that keratinocytes are valid models for comparing inhalation exposure. Although a single cell type cannot display the sensitivities of the numerous cell types that are exposed in the lung and elsewhere in the body, it permits calculating a benchmark suggestive of potential toxicity. We only offer a back-of-the-envelope calculation showing that the measured values are relevant. Extrapolating values based on "lung" cells are not necessarily more relevant to *in vivo* sensitivity. In view of all uncertainties, the estimate provided in the manuscript remains useful.

We added the following to Section 3.1., to offer perspective for extrapolating *in vitro* values:

"*Although no assay will successfully reproduce the* in vivo *environment, i.e., exposure to approximately 40 different cell types in the respiratory tract, distribution of toxicants throughout the bloodstream to major organs, and either deactivation or activation by metabolism, these (and other)* in vitro *results are useful as an estimation of risk.*"

(3)   The toxicity bioassay was quantified colorimetrically at the wavelength of 495 nm. This wavelength overlaps with the visible brown carbon absorption. Did the author measure the absorption from aerosol sample extracts to account for the potential interference from brown carbon absorption at this wavelength?

The envelope assay involves isolating these protein structures by washing them in SDS solution before measuring their protein content. Thus, the smoke extracts are separated from the envelopes, so no interference occurs from their color.

We added the following clarification to Section 2.4:
"*As the protein structures of SIK cultures exposed to WSA were isolated during the washing process, no interference from WSA absorption occurs in the colorimetric analysis.*"

(4)    Line 158: What were the positive controls used in the cell toxicity bioassay to simulate the maximum activity?

We added the following information to Section 2.4:
"*Ionophore X5375, which is known to induce envelope formation in the large majority of cells (Rice and Green, 1979), was used as the positive control.*"

(5)    Line 245: The cell toxicity measured by the protein content from envelope formation was not a direct measure of cell death. How did the authors determine whether the WSA extracts kill cells?

We have stated that envelope formation is actually a measure of cell death, since it occurs when the cells become permeabilized:

Line 143: "*The cross-linking can be induced by permeabilization of the cells, permitting a rise in cytosolic calcium ions that activates the membrane-bound transglutaminase TGM1 **and results in cell death**.*"

(6)    The rationale for measuring AhR and ER activity from woodsmoke aerosol exposure should be strengthen. What are the underlying hypotheses?

We added the following to the Introduction to strengthen our hypotheses:

"*We measured the total cellular toxicity effects of smoldering WSA. In addition, the Ah receptor (AhR) and estrogen receptor (ER) are known to play critical regulatory roles in mediating the biological and toxicological effects of diverse environmental chemicals. As wood combustion produces PAH-like compounds and phenolic compounds, classes of chemicals that are known to activate the AhR and/or ER signaling pathways (Denison and Nagy, 2003; Krüger et al., 2008; Stejskalova et al., 2011; Li et al., 2012; Cipolletti et al., 2018), we measured the AhR and ER activities of smoldering WSA*"

(7)    Throughout the manuscript, a lot of discussions between molecular composition and biological endpoints (e.g., AhR activity) were based on correlations. The authors should be careful about drawing conclusions from this relationship since correlation does not imply causation. Validation work should be carried out.

We agree that care should be taken when drawing conclusions from correlations. It is unreasonable, however, to avoid reaching some conclusions altogether from a correlation study. We have been careful to discuss findings in terms of what is known and what is speculative (e.g., "***potential chromophores***" (Line 69); "***it is not clear which compounds specifically effect** (sic) **cell toxicity***" (Line 346)) and we discussed at length how some of our findings are unexpected or may be coincidental. Here are examples of explicit caveats.

   a.   Line 323: "*A positive effect means that the signal of the peak directly …and a negative effect means anti-correlation, **neither of which is necessarily causative**.*" (this sentence is followed by direct examples of why correlation is not necessarily causation in our work)

b. Line 320: "*The occurrence of structural isomers is a limitation that may **confound the statistics** in this work…*"

c. Line 388: "*Even so, many of the correlations are not easily understood and **could be due to coincidence or driven by unassigned isomers***."

We mentioned that the correlation portion of this work "*identified multiple targets for further toxicological testing.*" Validation work should be done in future studies but is outside the scope of the current study.

(8)  For the brown carbon measurement, why did the author measure the visible light only (400-700 nm), but not the full range for tropospheric-relevant radiation (300-700 nm)? Note that most brown carbon constituents absorb light at near-UV wavelengths (300-400 nm) and to a lesser extent visible light.

We measured a broad UV-vis spectrum in the range of 200 – 800 nm, but only use the visible spectra to correlate with the HRMS data. We fully understand that BrC typically have high absorption at UV wavelengths and the corresponding author has previously published papers on BrC showing the full absorption spectra. However, this manuscript is focused only on understanding the molecular identities of the "brown" chromophores, i.e., "visible light absorption" is in the title and all of our MAC values are labeled as MACvis. As BrC is a mixture, there are many compounds in the aerosol that do not absorb light – in fact, it has been documented that the chromophoric fraction in some BrC is rather low but the chromophores are potent (Laskin et al., 2014). Doing the correlations with the inclusion of the UV portion of the absorption is expected to dilute the strength of the correlations to the visible-light-absorbing chromophores.

We added this sentence of clarification to Section 2.3. "*As this work is focused on visible-light-absorbing chromophores, the UV portion of the brown carbon absorption was not examined.*"

(9)  Line 380-385: What fractions of sinapaldehyde and coniferaldehyde converted to their enol forms in the extract solutions?

Sinapaldehyde and coniferaldehyde don't have accessible enol forms. In general, enol formation is highly unfavorable at atmospherically relevant temperatures (i.e., equilibrium is highly shifted toward keto form, e.g., (Keeffe et al., 1990), or organic chemistry textbooks), and for conjugated compounds like sinapaldehyde and coniferaldehyde, where a hypothetical enol formation removes aromaticity, this likelihood will be further reduced to zero. Our point is that with vanillydene acetone, this tautomerization is possible due to its neighboring sigma carbon. We made the following changes (underlined) to be more clear.

"*…but vanillylidene acetone can achieve some additional resonance with its keto-enol tautomerization (which tends not to happen in the ring because the aromatic phenol is highly favored) while sinapaldehyde and coniferaldehyde cannot.*"

(10)  Line 386: The authors could check on the pKa of these detected products and compare that to the pH of your solution.

We will clarify our discussion of the pH effects on line 386. The phenolates in question either occur either as isolated compounds in solution or as a member of a charge-transfer pair. As the pH of our solution is roughly 5, the free phenolate form of, say, coniferaldehyde (pKa ~ 8, Ragnar, Lindgren, and Nilvebrant, 2008) would be expected to be roughly 0.1% of the phenol form if equilibrium is reached in solution. However, as a member of a charge-transfer pair, the acid/base dissociation equilibrium is less important.  A charge transfer pair can exist at a wide variety of pH because of the localized effects.  Importantly, the pH-dependence of charge transfer pair to produce chromophores differs by their chemical structure. For example, the pH dependence of the chromophore from phenol + old yellow enzyme is strong for chlorophenol but weak for methoxybenzaldehyde (Abramovitz and Massey, 1976). Because of these differences, we stated in our conclusion that we "*hypothesize that the mechanism of action is through charge transfer reactions to form phenolates*."

We have now inserted the solution pH value and revised the text in question (revisions underlined) to be more clear:

"*It could be that Our solution may not be sufficiently basic (pH ~ 5) for the deprotonated form of some phenols to substantially exist in equilibrium (e.g., pKa for sinapaldehyde is 8.2, coniferaldehyde is 7.98, vanillin is 7.4, and bivanillin is 6.16,  (Ragnar et al., 2000); however, localized charge-transfer interactions may facilitate the deprotonation of phenols at pH much lower than their pKa (Abramovitz and Massey, 1976).*"

(11)  Line 423-425: Does the formation of phenolate occur only in the aqueous solution?

No, phenolates can be observed in aerosols as well (Phillips et al., 2017; Lin et al., 2017) as most aerosols have some water on them. Furthermore, on aerosols, there may not be just one pH but a distribution of pHs throughout the phase-separated pockets so the phenols can exist separately from phenolates.

"*In atmospheric aerosols, phenolates have been suggested to contribute substantially to brown carbon absorption as facilitated by their charge-transfer interactions (Phillips et al., 2017).*"

(12)  Line 12 and 86: did the author mean to say "hardwood" here?

No actually, we do mean "*heart*wood." The woody material of a tree can usually be divided into "sapwood" (less dense, living, higher-moisture, outermost portion of a woody stem or branch) and "heartwood" (more dense, dead, drier, inner wood). Usually the sapwood is a lighter color than the heartwood, and is named as such due to the high amount of sap in the cells. Heartwood often comprises the majority of the cross-section and is used to make furniture.

**Anonymous Referee #2**

This study investigated the molecular composition, optical and biological properties of woodsmoke aerosols. Correlations between some identified molecular formulas with MAC and bio endpoints were found. Overall, the analyses are sound, and the manuscript is well-written and easy to follow. I recommend that it can be published following some revisions.

(1)  I have the same concern as Referee#1 comments (1) and (2) for the selection of cell type. The author should illustrate the rationale to use human epidermal keratinocytes.

Please see the response to Referee #1 comments (1) and (2) above.

Also, why such high passage number (25-35) were being used? Usually the cell passage number shouldn't be higher than 20, otherwise the cell response results can be biased.

We are not aware of any specific limits on passage numbers of cell cultures in general (though they may be advocated for certain assays), but we do understand the concern about passage number. Since neoplastic cell lines evolve in culture, limits are commonly put on passage numbers to assure that the properties of the cells are not evolved far beyond those of the original passage. This is particularly germane if the passage numbers at the beginning of a study are quite different from those at the end. In this case we used passages of a narrow range to avoid such a problem. The SIK line is well characterized through numerous passages and is known to retain essentially normal function at the passage range employed. The cells have a single chromosomal aberration that permits continuous growth while altering their behavior only minimally (Rice et al, 1993). We observed that the cells were similarly sensitive to the measured effects of WSA exposure on envelope formation throughout the range of passages employed.

We added the following justification to Section 2.4:
*"SIK cells have a single chromosomal aberration that permits continuous growth throughout the range of passages employed while altering their behaviour only minimally (Rice et al, 1993)."*

(2)  Line 149: cells were incubated with extracts of WSA for 48h. During the 48h incubation in water/acetonitrile solution, cells may not be alive although the author did not find envelope formation in negative control. Did the authors try other cell viability assays to test cell viability? Why the author used such a long incubation time?

The long incubation time was used to increase the sensitivity of the assay when sample amounts were limited (thereby avoiding sample batch effects). Envelope formation is a measure of sufficient cell damage to induce permeabilization, a normal feature of their terminal differentiation (death). We did not test for lesser damage that might permit the cells to survive. Envelope formation is known to depend on transglutaminase activity, which is induced during differentiation. Although the less differentiated cells in the basal layer may lack sufficient transglutaminase to make envelopes, some 90% develop them in the post-confluent stratified cultures assayed (Rice and Green, 1979).

We added the following clarification to Section 2.4:
"*...to increase the assay sensitivity while conserving cell viability.*"

(3)  Line 263: no correlation between chromophoric species with cell toxicity does not necessarily mean they don't contribute to produce cell toxicity, and vice versa. Indeed, there is a recent paper (Chen et al., es&t 2019) showing significant correlation between DTT activity and water-soluble brown carbon.

We do not mean to imply that there is no correlation between aerosol chromophoric species and toxicity in general. We only reach this conclusion for the chromophores generated in our samples, as there are a vast variety of chromophores in nature that may produce a cell toxicity response (such as PAHs, nitrophenols, quinones, metal-ligand pairs). Chen et al (2019) tested WSOM extracts of ambient aerosols collected from China, where the findings were suggested by the authors to be due to the contributions from PAHs, quinones, or metals that generate ROS through redox cycling or Fenton-like chemistry, which are not studied in this work.

We added the Chen (2019) reference and clarify our statements by making the following changes (underlined):

"*Furthermore, the MACvis values do not correlate to either cell toxicity (**Fig. S5b**, adj. $R^2$ 0.03) or AhR activity (**Fig. S5c**, adj. $R^2$ -0.06), suggesting that chromophoric species in this work (e.g., specific conjugated compounds, charge transfer complexes, etc.) are not necessarily those producing these effects. In contrast, specific classes of chromophores from atmospheric aerosols in China have been found to have oxidative potential based on a chemical assay, which were suggested to be due to their PAH-like, quinone-like, or metallic constituents (Chen et al., 2019).*"

(Abstract) "*Finally, MACvis had no correlation with toxicity or receptor signaling, suggesting that key chromophores in this work are not biologically active on the endpoints tested.*"

(4)  There are a lot of correlation analysis in the manuscript. The authors should also do statistical analysis to show how significant the correlations are.

This is a good suggestion. In general, there is a relationship between the $R^2$ coefficient and the p value, an indicator of statistical significance. Although strict cutoffs are rarely used by statisticians, $p < 0.05$ is generally accepted as highly significant and for our data, the cutoff of $p = 0.05$ occurs around $R^2 = 0.44$ on average. We have done the significance analysis for all of the statistical fits in the present work using Matlab's correlation coefficient function and found that 90% of the entries in Table 2 had $p < 0.05$, which indicates strong evidence against the null hypothesis. The remainder were primarily borderline cases ($0.05 < p < 0.07$), which indicates weaker evidence. We made the following changes:

   a.  We added the following text to the caption of Table 2: "*Entries have p-values ≤ 0.05 (90%), or p < 0.1 (100%).*"

   b.  We added the following text to the Results and Discussion section of the manuscript: "*90% of the correlations shown in **Table 2** have p-values ranging from 0.0002 to 0.05, while 100%*

*have p < 0.1. This suggests that the correlations have moderate-to-strong evidence against coincidence.*"

(5)    The authors should do some comparisons of their cell toxicity, AhR activity and ER activity results with other previous studies using the same assays. Only looking at the numbers in table 1 cannot get a sense how toxic the WSA are. The authors shall compare their results with other types of aerosols or other toxicants.

Originally, we limited the discussion mainly to aerosols that were derived from woodsmoke; however, we can see the referee's point about the benefit of broader comparisons with other types of aerosols or other toxicants.

While we cannot compare the envelope-formation cell toxicity assay to other works because this is the first time this type of assay has been performed on atmospherically-relevant aerosols, we have added comparisons of AhR activity and ER activity for different size fractions of particulate matter (PM) to the Results and Discussion:

"*Despite dissimilarities in chemical composition, it can be instructive to compare the total AhR and ER activity of smoldering WSA to those measured for ambient particulate matter. Utilizing similar cell-based bioassays, $PM_1$ collected from rural and urban traffic sites in Switzerland was found to have AhR activity of $0.5 – 2 \times 10^{-6}$ g TCDD EQs/g PM and ER activity of $2 – 23 \times 10^{-9}$ g E2 EQs/g PM (Wenger et al., 2009a; Wenger et al., 2009b). Similarly, $PM_{10}$ collected from downtown Toronto, Canada was found to have AhR activity of $0.04 – 1 \times 10^{-6}$ g TCDD EQs/g PM and ER activity of $\sim 10^{-6}$ g E2 EQs/g PM (Clemons et al., 1998). PM of various size fractions from different polluted sites in the Czech Republic had AhR activity of $0.001 – 1 \times 10^{-6}$ g TCDD EQs/g PM and ER activity of $0.1 – 20 \times 10^{-9}$ g E2 EQs/g PM (Novák et al., 2014). Organic PM in Wuhan, China was found to have ER activity of $\sim 2 – 8 \times 10^{-7}$ g E2 EQs/g, depending on whether the PM was collected on sunny or foggy days (Wang et al., 2004). In comparison, the AhR activities of smoldering WSA in this work are within the range of PM from urban and industrial sites, while the ER activities are similar to some types of urban/industrial PM but a factor of $10^{2-3}$ lower than others. Wenger et al. (2009) also found the total concentration of AhR agonists to be much higher than concentrations for the traditional AhR agonists such as PCDD/Fs, suggesting a diversity of AhR-active ligands in ambient PM and further supporting the notion that AhR can bind and be activated by a wide range of structurally diverse chemicals in the environment (Denison and Nagy, 2003; DeGroot et al., 2011; Stejskalova et al., 2011).*"

(6)    Minor comment: 1. Figure S3, scale the x-axis to 0.1 to 4 mg/mL would be better to present the concentration-dependent curves.

We have changed the scale in the figure as recommended by the reviewer.

[revised manuscript text omitted]